# Disentangling Human Error from the Ground Truth in Segmentation of Medical Images

**Le Zhang**[1,2]*, **Ryutaro Tanno**[2,3]*, **Mou-Cheng Xu**[2], **Chen Jin**[2],
**Joseph Jacob**[2], **Olga Ciccarelli**[1], **Frederik Barkhof**[1,2] and **Daniel C. Alexander**[2]

[1]Queen Square Multiple Sclerosis Centre, Department of Neuroinflammation,
Queen Square Institute of Neurology, Faculty of Brain Sciences,
University College London, London, UK.
[2]Centre for Medical Image Computing, Department of Computer Science,
University College London, London, UK.
[3] Healthcare Intelligence, Microsoft Research, Cambridge, UK

le.zhang@ucl.ac.uk
rytanno@microsoft.com

## Abstract

Recent years have seen increasing use of supervised learning methods for segmentation tasks. However, the predictive performance of these algorithms depends on the quality of labels. This problem is particularly pertinent in the medical image domain, where both the annotation cost and inter-observer variability are high. In a typical label acquisition process, different human experts provide their estimates of the "true" segmentation labels under the influence of their own biases and competence levels. Treating these noisy labels blindly as the ground truth limits the performance that automatic segmentation algorithms can achieve. In this work, we present a method for jointly learning, from purely noisy observations alone, the reliability of individual annotators and the true segmentation label distributions, using two coupled CNNs. The separation of the two is achieved by encouraging the estimated annotators to be maximally unreliable while achieving high fidelity with the noisy training data. We first define a toy segmentation dataset based on MNIST and study the properties of the proposed algorithm. We then demonstrate the utility of the method on three public medical imaging segmentation datasets with simulated (when necessary) and real diverse annotations: 1) MSLSC (multiple-sclerosis lesions); 2) BraTS (brain tumours); 3) LIDC-IDRI (lung abnormalities). In all cases, our method outperforms competing methods and relevant baselines particularly in cases where the number of annotations is small and the amount of disagreement is large. The experiments also show strong ability to capture the complex spatial characteristics of annotators' mistakes. Our code is available at https://github.com/moucheng2017/Learn_Noisy_Labels_Medical_Images.

## 1 Introduction

Segmentation of anatomical structures in medical images is known to suffer from high inter-reader variability [1–5], influencing the performance of downstream supervised machine learning models. This problem is particularly prominent in the medical domain where the labelled data is commonly scarce due to the high cost of annotations. For instance, accurate identification of multiple sclerosis (MS) lesions in MRIs is difficult even for experienced experts due to variability in lesion location, size, shape and anatomical variability across patients [6]. Another example [4] reports the average

---

inter-reader variability in the range 74-85% for glioblastoma (a type of brain tumour) segmentation. Further aggravated by differences in biases and levels of expertise, segmentation annotations of structures in medical images suffer from high annotation variations [7]. In consequence, despite the present abundance of medical imaging data thanks to over two decades of digitisation, the world still remains relatively short of access to data with curated labels [8], that is amenable to machine learning, necessitating intelligent methods to learn robustly from such noisy annotations.

To mitigate inter-reader variations, different pre-processing techniques are commonly used to curate segmentation annotations by fusing labels from different experts. The most basic yet popular approach is based on the majority vote where the most representative opinion of the experts is treated as the ground truth (GT). A smarter version that accounts for similarity of classes has proven effective in aggregation of brain tumour segmentation labels [4]. A key limitation of such approaches, however, is that all experts are assumed to be equally reliable. Warfield *et al.*[9] proposed a label fusion method, called STAPLE that explicitly models the reliability of individual experts and uses that information to "weigh" their opinions in the label aggregation step. After consistent demonstration of its superiority over the standard majority-vote pre-processing in multiple applications, STAPLE has become the go-to label fusion method in the creation of public medical image segmentation datasets e.g., ISLES [10], MSSeg [11], Gleason'19 [12] datasets. Asman *et al.*later extended this approach in [13] by accounting for voxel-wise consensus to address the issue of under-estimation of annotators' reliability. In [14], another extension was proposed in order to model the reliability of annotators across different pixels in images. More recently, within the context of multi-atlas segmentation problems [15] where image registration is used to warp segments from labeled images ("atlases") onto a new scan, STAPLE has been enhanced in multiple ways to encode the information of the underlying images into the label aggregation process. A notable example is STEP proposed in Cardoso *et al.*[16] who designed a strategy to further incorporate the local morphological similarity between atlases and target images, and different extensions of this approach such as [17, 18] have since been considered. However, these previous label fusion approaches have a common drawback—they critically lack a mechanism to integrate information across different training images. This fundamentally limits the remit of applications to cases where each image comes with a reasonable number of annotations from multiple experts, which can be prohibitively expensive in practice. Moreover, relatively simplistic functions are used to model the relationship between observed noisy annotations, true labels and reliability of experts, which may fail to capture complex characteristics of human annotators.

In this work, we introduce the first instance of an end-to-end supervised segmentation method that jointly estimates, from noisy labels alone, the reliability of multiple human annotators and true segmentation labels. The proposed architecture (Fig. 1) consists of two coupled CNNs where one estimates the true segmentation probabilities and the other models the characteristics of individual annotators (e.g., tendency to over-segmentation, mix-up between different classes, etc) by estimating the pixel-wise confusion matrices (CMs) on a per image basis. Unlike STAPLE [9] and its variants, our method models, and disentangles with deep neural networks, the complex mappings from the input images to the annotator behaviours and to the true segmentation label. Furthermore, the parameters of the CNNs are "global variables" that are optimised across different image samples; this enables the model to disentangle robustly the annotators' mistakes and the true labels based on correlations between similar image samples, even when the number of available annotations is small per image (e.g., a single annotation per image). In contrast, this would not be possible with STAPLE [9] and its variants [14, 16] where the annotators' parameters are estimated on every target image separately.

For evaluation, we first simulate a diverse range of annotator types on the MNIST dataset by performing morphometric operations with Morpho-MNIST framework [19]. Then we demonstrate the potential in several real-world medical imaging datasets, namely (i) MS lesion segmentation dataset (MSLSC) from the ISBI 2015 challenge [20], (ii) Brain tumour segmentation dataset (BraTS) [4] and (iii) Lung nodule segmentation dataset (LIDC-IDRI) [21]. Experiments on all datasets demonstrate that our method consistently leads to better segmentation performance compared to widely adopted label-fusion methods and other relevant baselines, especially when the number of available labels for each image is low and the degree of annotator disagreement is high.

## 2   Related Work

The majority of algorithmic innovations in the space of *label aggregation for segmentation* have uniquely originated from the medical imaging community, partly due to the prominence of the

inter-reader variability problem in the field, and the wide-reaching values of reliable segmentation methods [14]. The aforementioned methods based on the STAPLE-framework such as [9, 13, 14, 16, 22, 17, 17, 18, 23] are based on generative models of human behaviours, where the latent variables of interest are the unobserved true labels and the "reliability" of the respective annotators. Our method can be viewed as an instance of translation of the STAPLE-framework to the supervised learning paradigm. As such, our method produces a model that can segment test images without needing to acquire labels from annotators or atlases unlike STAPLE and its local variants. Another key difference is that our method is jointly trained on many different subjects while the STAPLE-variants are only fitted on a per-subject basis. This means that our method is able to learn from correlations between different subjects, which previous works have not attempted— for example, our method uniquely can estimate the reliability and true labels even when there is only one label available per input image as shown later.

Our work also relates to a recent strand of methods that aim to generate a set of diverse and plausible segmentation proposals on a given image. Notably, probabilistic U-net [24] and its recent variants, PHiSeg [25] have shown that the aforementioned inter-reader variations in segmentation labels can be modelled with sophisticated forms of probabilistic CNNs. Such approaches, however, fundamentally differ from ours in that variable annotations from many experts in the training data are assumed to be all realistic instances of the true segmentation; we assume, on the other hand, that there is a single, unknown, true segmentation map of the underlying anatomy, and each individual annotator produces a noisy approximation to it with variations that reflect their individual characteristics. The latter assumption may be reasonable in the context of segmentation problems since there exists only one true boundary of the physical objects captured in an image while multiple hypothesis can arise from ambiguities in human interpretations.

We also note that, in standard classification problems, a plethora of different works have shown the utility of modelling the labeling process of human annotators in restoring the true label distribution [26–28]. Such approaches can be categorized into two groups: (1) *two-stage* approach [29–33], and (2) *simultaneous* approach [34–37, 27, 28, 38]. In the first category, the noisy labels are first curated through a probabilistic model of annotators, and subsequently, a supervised machine-learning model is trained on the curated labels. The initial attempt [29] was made in the early 1970s, and numerous advances such as [30–33] since built upon this work e.g. by estimating sample difficulty and human biases. In contrast, models in the second category aim to curate labels and learn a supervised model jointly in an end-to-end fashion [34–37, 27, 28] so that the two components inform each other. Although the evidence still remains limited to the simple classification task, these *simultaneous* approaches have shown promising improvements over the methods in the first category in terms of the predictive performance of the supervised model and the sample efficiency (i.e., fewer labels are required per input). However, to date very little attention has been paid to the same problem in more complicated, structured prediction tasks where the outputs are high dimensional. In this work, we propose the first *simultaneous* approach to addressing such a problem for image segmentation, while drawing inspirations from the STAPLE framework [9] which would fall into the *two-stage* approach category.

## 3 Method

### 3.1 Problem Set-up

In this work, we consider the problem of learning a supervised segmentation model from noisy labels acquired from multiple human annotators. Specifically, we consider a scenario where set of images $\{\mathbf{x}_n \in \mathbb{R}^{W \times H \times C}\}_{n=1}^{N}$ (with $W, H, C$ denoting the width, height and channels of the image) are assigned with noisy segmentation labels $\{\tilde{\mathbf{y}}_n^{(r)} \in \mathcal{Y}^{W \times H}\}_{n=1,...,N}^{r \in S(\mathbf{x}_i)}$ from multiple annotators where $\tilde{\mathbf{y}}_n^{(r)}$ denotes the label from annotator $r \in \{1,...,R\}$ and $S(\mathbf{x}_n)$ denotes the set of all annotators who labelled image $\mathbf{x}_i$ and $\mathcal{Y} = [1,2,...,L]$ denotes the set of classes.

Here we assume that every image $\mathbf{x}$ annotated by at least one person i.e., $|S(\mathbf{x})| \geq 1$, and no GT labels $\{\mathbf{y}_n \in \mathcal{Y}^{W \times H}\}_{n=1,...,N}$ are available. The problem of interest here is to *learn the unobserved true segmentation distribution $p(\mathbf{y} \mid \mathbf{x})$ from such noisy labelled dataset* $\mathcal{D} = \{\mathbf{x}_n, \tilde{\mathbf{y}}_n^{(r)}\}_{n=1,...,N}^{r \in S(\mathbf{x}_n)}$ i.e., the combination of images, noisy annotations and experts' identities for labels (which label was obtained from whom).

We also emphasise that *the goal at inference time is to segment a given unlabelled test image* but not to fuse multiple available labels as is typically done in multi-atlas segmentation approaches [15].

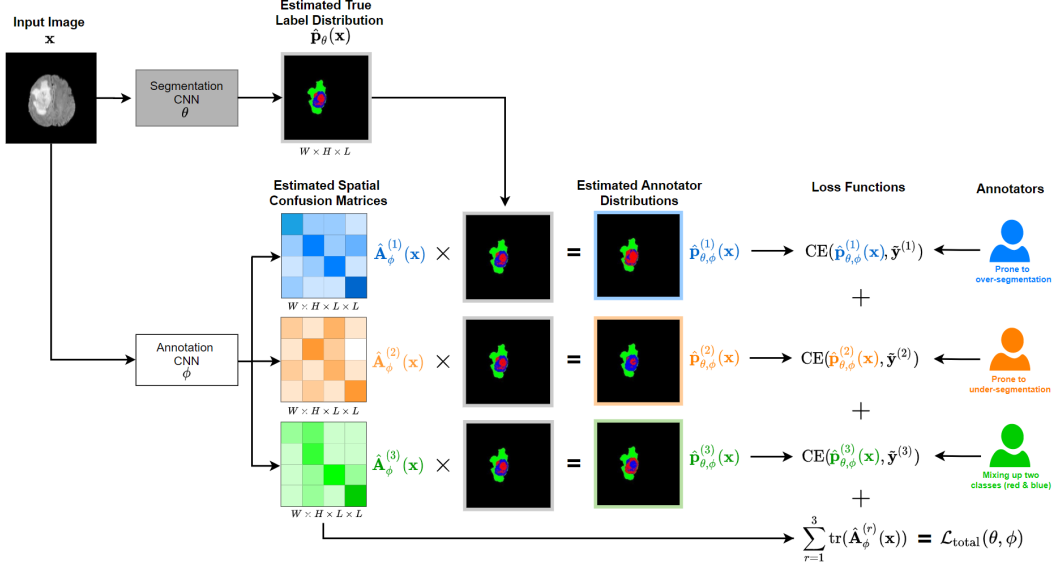

Figure 1: An architecture schematic in the presence of 3 annotators of varying characteristics (over-segmentation, under-segmentation and confusing between two classes, red and blue). The model consists of two parts: (1) *segmentation network* parametrised by $\theta$ that generates an estimate of the unobserved true segmentation probabilities, $\mathbf{p}_\theta(\mathbf{x})$; (2) *annotator network*, parametrised by $\phi$, that estimates the pixelwise confusion matrices (CMs), $\{\mathbf{A}_\phi^{(r)}(\mathbf{x})\}_{r=1}^3$ of the annotators for the given input image $\mathbf{x}$. During training, the estimated annotators distributions $\hat{\mathbf{p}}_{\theta,\phi}^{(r)}(\mathbf{x}) := \mathbf{A}_\phi^{(r)}(\mathbf{x}) \cdot \mathbf{p}_\theta(\mathbf{x})$ are computed, and the parameters $\{\theta,\phi\}$ are learned by minimizing the sum of their cross-entropy losses with respect to the acquired noisy segmentation labels $\tilde{\mathbf{y}}^{(r)}$, and the trace of the estimated CMs. At test time, the output of the segmentation network, $\hat{\mathbf{p}}_\theta(\mathbf{x})$ is used to yield the prediction.

## 3.2 Probabilistic Model and Proposed Architecture

Here we describe the probabilistic model of the observed noisy labels from multiple annotators. We make two key assumptions: (1) annotators are statistically independent, (2) annotations over different pixels are independent given the input image. Under these assumptions, the probability of observing noisy labels $\{\tilde{\mathbf{y}}^{(r)}\}_{r \in S(\mathbf{x})}$ on $\mathbf{x}$ factorises as:

$$p(\{\tilde{\mathbf{y}}^{(r)}\}_{r \in S(\mathbf{x})} \mid \mathbf{x}) = \prod_{r \in S(\mathbf{x})} p(\tilde{\mathbf{y}}^{(r)} \mid \mathbf{x}) = \prod_{\substack{r \in S(\mathbf{x}) \\ }} \prod_{\substack{w \in \{1,...,W\} \\ h \in \{1,...,H\}}} p(\tilde{y}_{wh}^{(r)} \mid \mathbf{x}) \quad (1)$$

where $\tilde{y}_{wh}^{(r)} \in [1,...,L]$ denotes the $(w,h)^{\text{th}}$ elements of $\tilde{\mathbf{y}}^{(r)} \in \mathcal{Y}^{W \times H}$. Now we rewrite the probability of observing each noisy label on each pixel $(w,h)$ as:

$$p(\tilde{y}_{wh}^{(r)} \mid \mathbf{x}) = \sum_{y_{wh}=1}^{L} p(\tilde{y}_{wh}^{(r)} \mid y_{wh}, \mathbf{x}) \cdot p(y_{wh} \mid \mathbf{x}) \quad (2)$$

where $p(y_{wh} \mid \mathbf{x})$ denotes the GT label distribution over the $(w,h)^{\text{th}}$ pixel in the image $\mathbf{x}$, and $p(\tilde{y}_{wh}^{(r)} \mid y_{wh}, \mathbf{x})$ describes the noisy labelling process by which annotator $r$ corrupts the true segmentation label. In particular, we refer to the $L \times L$ matrix whose each $(i,j)^{\text{th}}$ element is defined by the second term $\mathbf{a}^{(r)}(\mathbf{x},w,h)_{ij} := p(\tilde{y}_{wh}^{(r)} = i \mid y_{wh} = j, \mathbf{x})$ as the CM of annotator $r$ at pixel $(w,h)$ in image $\mathbf{x}$.

We introduce a CNN-based architecture which models the different constituents in the above joint probability distribution $p(\{\tilde{\mathbf{y}}^{(r)}\}_{r \in S(\mathbf{x})} \mid \mathbf{x})$ as illustrated in Fig. 1. The model consists of two components: (1) *Segmentation Network*, parametrised by $\theta$, which estimates the GT segmentation probability map, $\hat{\mathbf{p}}_\theta(\mathbf{x}) \in \mathbb{R}^{W \times H \times L}$ whose each $(w,h,i)^{\text{th}}$ element approximates $p(y_{wh} = i \mid \mathbf{x})$;(2) *Annotator Network*, parametrised by $\phi$, that generate estimates of the pixel-wise CMs of respective annotators as a function of the input image, $\{\hat{\mathbf{A}}_\phi^{(r)}(\mathbf{x}) \in [0,1]^{W \times H \times L \times L}\}_{r=1}^R$ whose each $(w,h,i,j)^{\text{th}}$ element approximates $p(\tilde{y}_{wh}^{(r)} = i \mid y_{wh} = j, \mathbf{x})$. Each product $\hat{\mathbf{p}}_{\theta,\phi}^{(r)}(\mathbf{x}) := \hat{\mathbf{A}}_\phi^{(r)}(\mathbf{x}) \cdot \hat{\mathbf{p}}_\theta(\mathbf{x})$ represents the

estimated segmentation probability map of the corresponding annotator. Note that here "·" denotes the element-wise matrix multiplications in the spatial dimensions $W, H$. At inference time, we use the output of the segmentation network $\hat{\mathbf{p}}_\theta(\mathbf{x})$ to segment test images.

We note that each spatial CM, $\hat{\mathbf{A}}_\phi^{(r)}(\mathbf{x})$ contains $WHL^2$ variables, and calculating the corresponding annotator's prediction $\hat{\mathbf{p}}_{\theta,\phi}^{(r)}(\mathbf{x})$ requires $WH(2L-1)L$ floating-point operations, potentially incurring a large time/space cost when the number of classes is large. Although not the focus of this work (as we are concerned with medical imaging applications for which the number of classes are mostly limited to less than 10), we also consider a low-rank approximation (rank=1) scheme to alleviate this issue wherever appropriate. More details are provided in the supplementary.

## 3.3 Learning Spatial Confusion Matrices and True Segmentation

Next, we describe how we jointly optimise the parameters of segmentation network, $\theta$ and the parameters of annotator network, $\phi$. In short, we minimise the negative log-likelihood of the probabilistic model plus a regularisation term via stochastic gradient descent. A detailed description is provided below.

Given training input $\mathbf{X} = \{\mathbf{x}_n\}_{n=1}^N$ and noisy labels $\tilde{\mathbf{Y}}^{(r)} = \{\tilde{\mathbf{y}}_n^{(r)} : r \in S(\mathbf{x}_n)\}_{n=1}^N$ for $r = 1,...,R$, we optimaize the parameters $\{\theta, \phi\}$ by minimizing the negative log-likelihood (NLL), $-\log p(\tilde{\mathbf{Y}}^{(1)},...,\tilde{\mathbf{Y}}^{(R)}|\mathbf{X})$. From eqs. (1) and (2), this optimization objective equates to the sum of cross-entropy losses between the observed noisy segmentations and the estimated annotator label distributions:

$$-\log p(\tilde{\mathbf{Y}}^{(1)},...,\tilde{\mathbf{Y}}^{(R)}|\mathbf{X}) = \sum_{n=1}^N \sum_{r=1}^R \mathbb{1}(r \in \mathcal{S}(\mathbf{x}_n)) \cdot \mathrm{CE}(\hat{\mathbf{A}}_\phi^{(r)}(\mathbf{x}_n) \cdot \hat{\mathbf{p}}_\theta(\mathbf{x}_n),\ \tilde{\mathbf{y}}_n^{(r)}) \tag{3}$$

Minimizing the above encourages each annotator-specific predictions $\hat{\mathbf{p}}_{\theta,\phi}^{(r)}(\mathbf{x})$ to be as close as possible to the true noisy label distribution of the annotator $\mathbf{p}^{(r)}(\mathbf{x})$. However, this loss function alone is not capable of separating the annotation noise from the true label distribution; there are many combinations of pairs $\hat{\mathbf{A}}_\phi^{(r)}(\mathbf{x})$ and segmentation model $\hat{\mathbf{p}}_\theta(\mathbf{x})$ such that $\hat{\mathbf{p}}_{\theta,\phi}^{(r)}(\mathbf{x})$ perfectly matches the true annotator's distribution $\mathbf{p}^{(r)}(\mathbf{x})$ for any input $\mathbf{x}$ (e.g., permutations of rows in the CMs). To combat this problem, inspired by Tanno *et al.*[28], which addressed an analogous issue for the classification task, we add the trace of the estimated CMs to the loss function in Eq. (3) as a regularisation term (see Sec 3.4). We thus optimize the combined loss:

$$\mathcal{L}_{\text{total}}(\theta, \phi) := \sum_{n=1}^N \sum_{r=1}^R \mathbb{1}(r \in \mathcal{S}(\mathbf{x}_n)) \cdot \left[ \mathrm{CE}(\hat{\mathbf{A}}_\phi^{(r)}(\mathbf{x}_n) \cdot \hat{\mathbf{p}}_\theta(\mathbf{x}_n),\ \tilde{\mathbf{y}}_n^{(r)}) + \lambda \cdot \mathrm{tr}(\hat{\mathbf{A}}_\phi^{(r)}(\mathbf{x}_n)) \right] \tag{4}$$

where $\mathcal{S}(\mathbf{x})$) denotes the set of all labels available for image $\mathbf{x}$, and $\mathrm{tr}(\mathbf{A})$ denotes the trace of matrix $\mathbf{A}$. The mean trace represents the average probability that a randomly selected annotator provides an accurate label. Intuitively, minimising the trace encourages the estimated annotators to be maximally unreliable while minimising the cross entropy ensures fidelity with observed noisy annotators. We minimise this combined loss via stochastic gradient descent to learn both $\{\theta, \phi\}$.

## 3.4 Justification for the Trace Norm

Here we provide a further justification for using the trace regularisation. Tanno *et al.*[28] showed that if the average CM of annotators is *diagonally dominant*, and the cross-entropy term in the loss function is zero, minimising the trace of the estimated CMs uniquely recovers the true CMs. However, their results concern properties of the average CMs of both the annotators and the classifier over the data population, rather than individual data samples. We show a similar but slightly weaker result in the sample-specific regime, which is more relevant as we estimate CMs of respective annotators on every input image.

First, let us set up the notations. For brevity, for a given input image $\mathbf{x} \in \mathbb{R}^{W \times H \times C}$, we denote the ground-truth CM of annotator $r$ at $(i,j)^{\text{th}}$ pixel and its estimate by $\mathbf{A}^{(r)} := [\mathbf{A}^{(r)}(\mathbf{x})_{ij}]$ and $\hat{\mathbf{A}}^{(r)} := [\hat{\mathbf{A}}^{(r)}(\mathbf{x})_{ij}] \in [0,1]^{L \times L}$, respectively. We also define the mean CM $\mathbf{A}^* := \sum_{r=1}^R \pi_r \mathbf{A}^{(r)}$ and its estimate $\hat{\mathbf{A}}^* := \sum_{r=1}^R \pi_r \hat{\mathbf{A}}^{(r)}$ where $\pi_r \in [0,1]$ is the probability that the annotator $r$ labels image $\mathbf{x}$. Lastly, as we stated earlier, we assume there is a single GT segmentation label per image — thus the

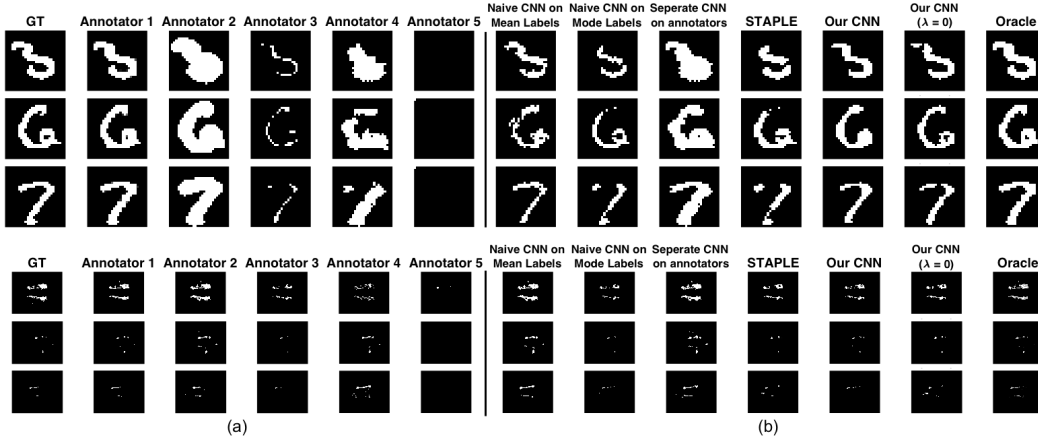

Figure 2: Visualisation of segmentation labels on two datasets: (a) ground-truth (GT) and segmentation labels from simulated annotators (Annotators 1 - 5); (b) the predictions from the supervised models.

true $L$-dimensional probability vector at pixel $(i,j)$ takes the form of a one-hot vector i.e., $\mathbf{p}(\mathbf{x}) = \mathbf{e}_k$ for, say, class $k \in [1,...,L]$. Then, the followings result motivates the use of the trace regularisation:

**Theorem 1.** *If the annotator's segmentation probabilities are perfectly modelled by the model for the given image i.e., $\hat{\mathbf{A}}^{(r)} \hat{\mathbf{p}}_\theta(\mathbf{x}) = \mathbf{A}^{(r)} \mathbf{p}(\mathbf{x}) \forall r = 1,...,R$, and the average true confusion matrix $\mathbf{A}^*$ at a given pixel and its estimate $\hat{\mathbf{A}}^*$ satisfy that $a_{kk}^* > a_{kj}^*$ for $j \neq k$ and $\hat{a}_{ii}^* > \hat{a}_{ij}^*$ for all $i,j$ such that $j \neq i$, then $\mathbf{A}^{(1)},...,\mathbf{A}^{(R)} = argmin_{\hat{\mathbf{A}}^{(1)},...,\hat{\mathbf{A}}^{(R)}} \left[ tr(\hat{\mathbf{A}}^*) \right]$ and such solutions are **unique** in the $k^{th}$ column where $k$ is the correct pixel class.*

The corresponding proof is provided in the supplementary material. The above result shows that if each estimated annotator's distribution $\hat{\mathbf{A}}^{(r)} \hat{\mathbf{p}}_\theta(\mathbf{x})$ is very close to the true noisy distribution $\mathbf{p}^{(r)}(\mathbf{x})$ (which is encouraged by minimizing the cross-entropy loss), and for a given pixel, the average CM has the $k^{th}$ diagonal entry larger than any other entries in the same row [2], then minimizing its trace will drive the estimates of the $k^{th}$ ('correct class') columns in the respective annotator's CMs to match the true values. Although this result is weaker than what was shown in [28] for the population setting rather than the individual samples, the single-ground-truth assumption means that the remaining values of the CMs are uniformly equal to $1/L$, and thus it suffices to recover the column of the correct class.

To encourage $\{\hat{\mathbf{A}}^{(1)},...,\hat{\mathbf{A}}^{(R)}\}$ to be also diagonally dominant, we initialize them with identity matrices by training the *annotation network* to maximise the trace for sufficient iterations as a warm-up period. Intuitively, the combination of the trace term and cross-entropy separates the true distribution from the annotation noise by finding the maximal amount of confusion which explains the noisy observations well.

## 4    Experiments

We evaluate our method on a variety of datasets including both synthetic and real-world scenarios:1) for MNIST segmentation and ISBI2015 MS lesion segmentation challenge dataset [39], we apply morphological operations to generate synthetic noisy labels in binary segmentation tasks; 2) for BraTS 2019 dataset [4], we apply similar simulation to create noisy labels in a multi-class segmentation task; 3) we also consider the LIDC-IDRI dataset which contains multiple annotations per input acquired from different clinical experts as the evaluation in practice. The etails of noisy label simulation can be found in Appendix A.1.

Our experiments are based on the assumption that no ground-truth (GT) label is not known a priori, hence, we compare our method against multiple label fusion methods. In particular, we consider four label fusion baselines: a) mean of all of the noisy labels; b) mode labels by taking the "majority vote"; c) label fusion via the original STAPLE method [9]; d) Spatial STAPLE, a more recent extension

of c) that accounts for spatial variations in CMs. After curating the noisy annotations via the above methods, we train the segmentation network and report the results. For c) and d), we used the toolkit[3]. To get an upper-bound performance, we also include the *oracle* model that is directly trained on the ground-truth annotations. To test the value of the proposed image-dependent spatial CMs, we also include "Global CM" model where a single CM is learned per annotator but fixed across pixels and images (analogous to *et al.*[34, 27, 28], but in segmentation task). Lastly, we also compare against a recent method called Probabilistic U-net as another baseline, which has been shown to capture inter-reader variations accurately. The details are presented in Appendix A.2.

For evaluation metrics, we use: 1) root-MSE between estimated CMs and real CMs; 2) Dice coefficient (DICE) between estimated segmentation and true segmentation; 3) The generalized energy distance proposed in [24] to measure the quality of the estimated annotator's labels.

## 4.1 MNIST and MS lesion segmentation datasets

MNIST dataset consists of 60,000 training and 10,000 testing examples, all of which are $28 \times 28$ grayscale images of digits from 0 to 9, and we derive the segmentation labels by thresholding the intensity values at 0.5. The MS dataset is publicly available and comprises 21 3D scans from 5 subjects. All scans are split into 10 for training and 11 for testing. We hold out 20% of training images as a validation set for both datasets. On both datasets, our proposed model achieves a higher dice similarity coefficient than STAPLE on the dense label case and, even more prominently, on the single label (i.e., randomly choose 1 label per image, aka, "one label per image") case (shown in Tables. 1&2 and Fig. 2). In addition, our model outperforms STAPLE without or with trace norm, in terms of CM estimation, specifically, we could achieve an increase at 6.3%. Additionally, we include the performance on different regularisation coefficient, which is presented in Fig. 3. Fig. 4 compares the segmentation accuracy on MNIST and MS lesion for a range of average dice where labels are generated by a group of 5 simulated annotators. Fig. 5 illustrates our model can capture the patterns of mistakes for each annotator. We also notice that our model is consistently more accurate than the global CM model, indicating the value of image-dependent pixel-wise CMs.

| Models | MNIST DICE (%) | MNIST CM estimation | MSLesion DICE (%) | MSLesion CM estimation |
|---|---|---|---|---|
| Naive CNN on mean labels | $38.36 \pm 0.41$ | n/a | $46.55 \pm 0.53$ | n/a |
| Naive CNN on mode labels | $62.89 \pm 0.63$ | n/a | $47.82 \pm 0.76$ | n/a |
| Probabilistic U-net [24] | $65.12 \pm 0.83$ | n/a | $46.15 \pm 0.59$ | n/a |
| Separate CNNs on annotators | $70.44 \pm 0.65$ | n/a | $46.84 \pm 1.24$ | n/a |
| STAPLE [9] | $78.03 \pm 0.29$ | $0.1241 \pm 0.0011$ | $55.05 \pm 0.53$ | $0.1502 \pm 0.0026$ |
| Spatial STAPLE [14] | $78.96 \pm 0.22$ | $0.1195 \pm 0.0013$ | $58.37 \pm 0.47$ | $0.1483 \pm 0.0031$ |
| Ours with Global CMs | $79.21 \pm 0.41$ | $0.1132 \pm 0.0028$ | $61.58 \pm 0.59$ | $0.1449 \pm 0.0051$ |
| Ours without Trace | $79.63 \pm 0.53$ | $0.1125 \pm 0.0037$ | $65.77 \pm 0.62$ | $0.1342 \pm 0.0053$ |
| Ours | $82.92 \pm 0.19$ | $0.0893 \pm 0.0009$ | $67.55 \pm 0.31$ | $0.0811 \pm 0.0024$ |
| Oracle (Ours but with known CMs) | $83.29 \pm 0.11$ | $0.0238 \pm 0.0005$ | $78.86 \pm 0.14$ | $0.0415 \pm 0.0017$ |

Table 1: Comparison of segmentation accuracy (DICE) and quality of confusion matrix (CM) estimation (MSE) for different methods with dense labels (mean $\pm$ standard deviation).

| Models | MNIST DICE (%) | MNIST CM estimation | MSLesion DICE (%) | MSLesion CM estimation |
|---|---|---|---|---|
| Naive CNN | $32.79 \pm 1.13$ | n/a | $27.41 \pm 1.45$ | n/a |
| STAPLE [9] | $54.07 \pm 0.68$ | $0.2617 \pm 0.0064$ | $35.74 \pm 0.84$ | $0.2833 \pm 0.0081$ |
| Spatial STAPLE [14] | $56.73 \pm 0.53$ | $0.2384 \pm 0.0061$ | $38.21 \pm 0.71$ | $0.2591 \pm 0.0074$ |
| Ours with Global CMs | $59.01 \pm 0.65$ | $0.1953 \pm 0.0041$ | $40.32 \pm 0.68$ | $0.1974 \pm 0.0063$ |
| Ours without Trace | $74.48 \pm 0.37$ | $0.1538 \pm 0.0029$ | $54.76 \pm 0.66$ | $0.1745 \pm 0.0044$ |
| Ours | $76.48 \pm 0.25$ | $0.1329 \pm 0.0012$ | $56.43 \pm 0.47$ | $0.1542 \pm 0.0023$ |

Table 2: Comparison of segmentation accuracy (DICE) and error of CM estimation (MSE) for different methods with one label per image (mean $\pm$ standard deviation). We note that 'Naive CNN' is a baseline trained by simply minimising the cross-entropy between the predictions and the noisy labels.

| Models | MNIST | MS | BraTS | LIDC-IDRI |
|---|---|---|---|---|
| Probabilistic U-net [24] | $1.46 \pm 0.04$ | $1.91 \pm 0.03$ | $3.23 \pm 0.07$ | $1.97 \pm 0.03$ |
| Ours | $\mathbf{1.24 \pm 0.02}$ | $\mathbf{1.67 \pm 0.03}$ | $\mathbf{3.14 \pm 0.05}$ | $\mathbf{1.87 \pm 0.04}$ |

Table 3: Comparison of Generalised Energy Distance (GED) on different datasets (mean $\pm$ standard deviation). The distance metric used here is the DICE score.

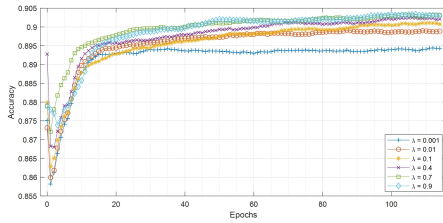

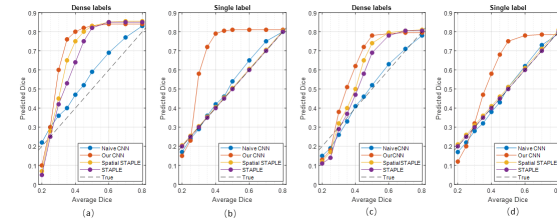

Figure 3: Curves of validation accuracy during training of our model for a range of hyperparameters. For our method, the scaling of trace regularizer is varied in [0.001, 0.01, 0.1, 0.4, 0.7, 0.9].)

Figure 4: Segmentation accuracy of different models on MNIST (a, b) and MS (c, d) dataset for a range of annotation noise levels (measured in average DICE score with respect to the GT labels.

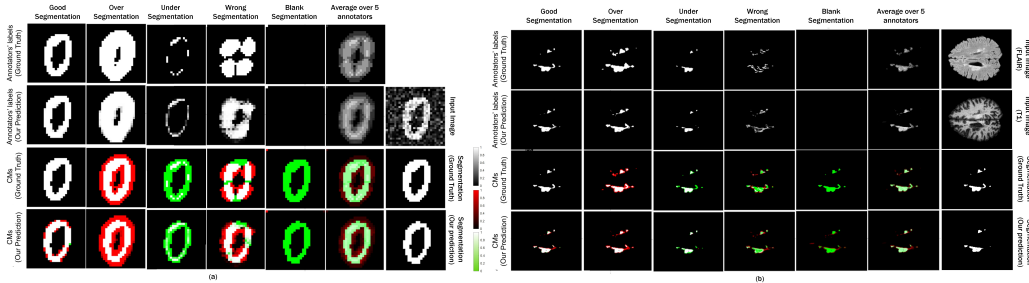

Figure 5: Visualisation of the estimated true labels and the estimated pixel-wise confusion matrices on MNIST/MS datasets along with their targets (best viewed in colour). White is the true positive, green is the false negative, red is the false positive and black is the true negative.

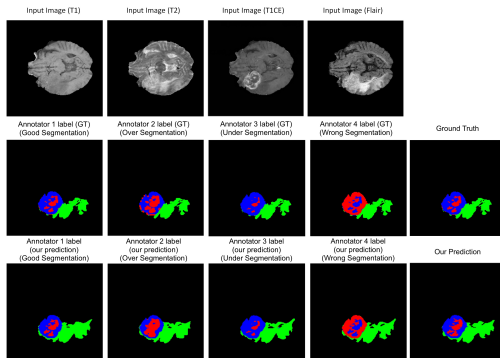

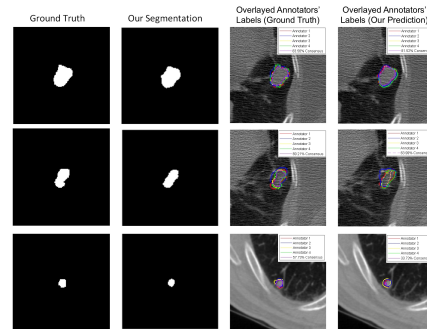

Figure 6: Visualisation of the estimated true segmentation on the BraTS dataset and the estimated annotations of their respective annotators (best viewed in colour). The tumour core (red) is the target class on which annotation mistakes are simulated.

Figure 7: Segmentation results on LIDC-IDRI dataset and the visualization of each annotator contours and the consensus. The bottom row shows an interesting example in which annotator 4 (green) misses the abnormality completely, which is also predicted by our model.

## 4.2 BraTS Dataset and LIDC-IDRI Dataset

We also evaluate our model on a multi-class segmentation task, using all of the 259 high grade glioma (HGG) cases in training data from 2019 multi-modal Brain Tumour Segmentation Challenge (BraTS). We extract each slice as 2D images and split them at case-wise to have, 1600 images for training, 300 for validation and 500 for testing. Pre-processing includes: concatenation of all of available modalities; centre cropping to 192 x 192; normalisation for each case at each modality. To create synthetic noisy labels in multi-class scenario, we first choose a target class and then apply morphological operations on the provided GT mask to create 4 synthetic noisy labels at different patterns, namely, over-segmentation, under-segmentation, wrong segmentation and good segmentation. The details of noisy label simulation are in Appendix A.3.

Lastly, we use the LIDC-IDRI dataset to evaluate our method in the scenario where multiple labels are acquired from different clinical experts. The dataset contains 1018 lung CT scans from 1010 lung patients with manual lesion segmentations from four experts. For each scan, 4 radiologists provided

annotation masks for lesions that they independently detected and considered to be abnormal. For our experiments, we used the same method in [24] to pre-process all scans. We split the dataset at case-wise into a training (722 patients), validation (144 patients) and testing (144 patients). We then resampled the CT scans to $1mm \times 1mm$ in-plane resolution. We also centre cropped 2D images ($180 \times 180$ pixels) around lesion positions, in order to focus on the annotated lesions. The lesion positions are those where at least one of the experts segmented a lesion. We hold 5000 images in the training set, 1000 images in the validation set and 1000 images in the test set. Since the dataset does not provide a single curated ground-truth for each image, we created a "gold standard" by aggregating the labels via STAPLE [14], a recent variant of the STAPLE framework employed in the creation of public medical image segmentation datasets e.g., ISLES [10], MSSeg [11], Gleason'19 [12] datasets. We further note that, as before, we assume labels are only available to the model during training, but not at test time, thus label aggregation methods cannot be applied on the test examples.

On both BraTS and LIDC-IDRI datasets, our proposed model achieves a higher dice similarity coefficient than STAPLE and Spatial STAPLE on both of the dense labels and single label scenarios (shown in Table. 4 and Table. 5 in Appendix A.3). In addition, our model (with trace) outperforms STAPLE in terms of CM estimation by a large margin at $14.4\%$ on BraTS. In Fig. 6, we visualized the segmentation results on BraTS and the corresponding annotators' predictions. Fig. 7 presents three examples of the segmentation results and the corresponding four annotator contours, as well as the consensus. As shown in both figures, our model successfully predicts both the segmentation of lesions and the variations of each annotator in different cases. We also measure the inter-reader consensus levels by computing the IoU of multiple annotations, and compare the segmentation performance in three subgroups of different consensus levels (low, medium and high). Results are shown in Fig. 14 and Fig. 15 in Appendix A.3.

Additionally, as shown in Table.3, our model consistently outperforms Probabilistic U-Net on generalized energy distance across the four test different datasets, indicating our method can better capture the inter-annotator variations than the baseline Probabilistic U-Net. This result shows that the information about which labels are acquired from whom is useful in modelling the variability in the observed segmentation labels.

## 5   Discussion and Conclusion

We introduced the first supervised segmentation method for jointly estimating the spatial characteristics of labelling errors from multiple human annotators and the ground-truth label distribution. We demonstrated this method on real-world datasets with both synthetic and real-world annotations. Our method is capable of estimating individual annotators and thereby improving robustness against label noise. Experiments have shown our model achieves considerable improvement over the traditional label fusion approaches including averaging, the majority vote and the widely used STAPLE framework and its recent extensions, in terms of both segmentation accuracy and the quality of confusion matrix (CM) estimation.

In the future, we aim to extend this work both in theory and applications. Here we made a simplifying assumption that there is a single, unknown, true segmentation map of the underlying anatomy, and each individual annotator produces a noisy approximation to it with variations that reflect their individual characteristics. This is in stark contrast with many recent advances (e.g., Probabilistic U-net [24] and PHiSeg [25]) that assume variable annotations from experts are all realistic instances of the true segmentation. One could argue that single-truth assumption may be sensible in the context of segmentation problems since there exists only one true boundary of the physical objects captured in an image while multiple hypothesis can arise from ambiguities in human interpretations. However, we believe that the reality lies somewhere between i.e., some variations are indeed intrinsic while some are specific to human imperfections. Separation of the two could be potentially addressed by using some prior knowledge about the individual annotators (e.g., meta-information such as the years of experiences, etc) [34] or using a small portion of dataset with curated annotations as a reference set which can be assumed to come from the true label distribution.

Another exciting avenue of research is the application of the annotation models in downstream tasks. Of particular interest is the design of active data collection schemes where the segmentation model is used to select which samples to annotate ("active learning"), and the annotator models are used to decide which experts to label them ("active labelling")—e.g., extending Yan *et al.*[40] from simple classification task to segmentation remains future work. Another exciting application is education of inexperienced annotators; the estimated spatial characteristics of segmentation mistakes provide further insights into their annotation behaviours, and as a result, potential help them improve the quality of their annotations in the next data acquisition.

## Boarder Impact Statement

*Image segmentation* has been one of the main challenges in modern medical image analysis, and describes the process of assigning each pixel or voxel in images with biologically meaningful discrete labels, such as anatomical structures and tissue types (e.g. pathology and healthy tissues). The task is required in many clinical and research applications, including surgical planning [41, 42], and the study of disease progression, aging or healthy development [43–45]. However, there are many cases in practice where the correct delineation of structures is challenging; this is also reflected in the well-known presence of high inter- and intra-reader variability in segmentation labels obtained from trained experts [9, 23, 5].

Although expert manual annotations of lesions is feasible in practice, this task is time consuming. It usually takes 1.5 to 2 hours to label a MS patient with average 3 visit scans. Meanwhile, the long-established gold standard for segmentation of medical images has been manually voxel-by-voxel labeled by an expert anatomist. Unfortunately, this process is fraught with both interand intra-rater variability (e.g., on the order of approximately 10% by volume [46, 47]). Thus, developing an automatic segmentation technique to fix the variability among inter- and intra-readers could be meaningful not only in terms of the accuracy in delineating MS lesions but also in the related reductions in time and economic costs derived from manual lesion labeling. The lack of consistency in labelling is also common to see in other medical imaging applications, e.g., in lung abnormalities segmentation from CT images. A lesion might be clearly visible by one annotator, but the information about whether it is cancer tissue or not might not be clear to others. While our work in the current form has only been demonstrated on medical images, we would like to stress that the medical imaging domain offers a considerably broad range of opportunities for impact; e.g., diagnosis/prognosis in radiology, surgical planning and study of disease progression and treatment, etc. In addition, the annotator information could be potentially utilised for the purpose of education. Another potential opportunity is to integrate such information into the data/label acquisition scheme in order to train reliable segmentation algorithms in a data-efficient manner.

## Acknowledgement and Funding Disclosure

We would like to thank Swami Sankaranarayanan and Ardavan Saeedi at Butterfly Network for their feedback and initial discussions. Mou-Cheng is supported by GSK funding (BIDS3000034123) via UCL EPSRC CDT in i4health and UCL Engineering Dean's Prize. We are also grateful for EPSRC grants EP/R006032/1, EP/M020533/1, CRUK/EPSRC grant NS/A000069/1, and the NIHR UCLH Biomedical Research Centre, which support this work.

## Footnotes

[2]For the standard "majority vote" label to capture the correct true labels, one requires the $k^{th}$ diagonal element in the average CM to be larger than the sum of the remaining elements in the same row, which is a more strict condition.

[3]https://www.nitrc.org/projects/masi-fusion/

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
