[Supplementary Material]

# Supplementary Material:
# Disentangling Human Error from Ground Truth in Segmentation of Medical Images

## A    Additional results

### A.1    Annotation Simulation Details

We generate synthetic annotations from an assumed GT on MNIST, MS lesion and BraTS datasets, to generate efficacy of the approach in an idealised situation where the GT is known. We simulate a group of 5 annotators of disparate characteristics by performing morphological transformations (e.g., thinning, thickening, fractures, etc) on the ground-truth (GT) segmentation labels, using Morpho-MNIST software [19]. In particular, the first annotator provides faithful segmentation ("good-segmentation") with approximate GT, the second tends over-segment ("over-segmentation"), the third tends to under-segment ("under-segmentation"), the fourth is prone to the combination of small fractures and over-segmentation ("wrong-segmentation") and the fifth always annotates everything as the background ("blank-segmentation"). To create synthetic noisy labels in multi-class scenario, we first choose a target class and then apply morphological operations on the provided GT mask to create 4 synthetic noisy labels at different patterns, namely, over-segmentation, under-segmentation, wrong segmentation and good segmentation. We create training data by deriving labels from the simulated annotators. We also experimented with varying the levels of morphological operations on MNIST and MS lesion datasets, to test the robustness of our methods to varying degrees of annotation noise.

### A.2    Additional Qualitative Results on MNIST and MS Dataset

Here we provide additional qualitative comparison of segmentation results and CM visualization results on MNIST and MS datasets. We examine the ability of our method to learn the CMs of annotators and the true label distribution on single label per image. Fig. 7 and Fig. 9 show the segmentation results on MNIST dataset on single label per image. Our model achieved a higher dice similarity coefficient than STAPLE and Spatial STAPLE, even prominently, our model outperformed STAPLE and Spatial STAPLE without or with trace norm, in terms of CM estimation. Fig. 8 and Fig. 10 illustrate our model on single label still can capture the patterns of mistakes.

Figure 7: Visualisation of segmentation labels on MNIST dataset for single label per image: (a) GT and simulated annotator's segmentations (Annotator 1 - 5); (b) the predictions from the supervised models.

Figure 8: Visualisation of estimated true labels and confusion matrices for single label per image on MNIST datasets (Best viewed in colour: white is the true positive, green is the false negative, red is the false positive and black is the true negative)

.

Figure 9: Visualisation of segmentation labels on MS lesion dataset for single label per image: (a) GT and simulated annotator's segmentations (Annotator 1 - 5); (b) the predictions from the supervised models.

Figure 10: Visualisation of estimated true labels and confusion matrices for single label per image on MS lesion datasets (Best viewed in colour: white is the true positive, green is the false negative, red is the false positive and black is the true negative).

### A.3  Quantitative and Extra Qualitative Results on BraTS and LIDC-IDRI

Here we provide the quantitative comparison of our method and other baselines on BraTS and LIDC-IDRI datasets, which have been precluded from the main text due to the space limit (see Table. 4 and Table. 5). We also provide additional qualitative examples (see Fig. 11,12, 13) on both datasets. Lastly, we compare the segmentation performance on 3 different subgroups of LIDC-IDRI with varying levels of inter-reader variability; Fig. 15 illustrates our method attains consistent improvement over the baselines in all cases, indicating its ability to segment more robustly even the hard examples where the experts in reality have disagreed to a large extent.

BraTS 2019 is a multi-class segmentation dataset, containing 259 cases with high grade (HG) and 76 cases with low grade (LG) glioma (a type of brain tumour). For each case, four MRI modalities are available, FLAIR, T1, T1-contrast and T2. The datasets are pre-processed by the organizers and co-registered to the same anatomical template, interpolated to the same resolution ($1\ mm^3$) and skull-stripped. We used only high grade cases and centre cropped 2D images ($192 \times 192$ pixels) and hold 1600 2D images for training, 300 images for validation, 500 images for testing, we apply Gaussian normalization on each case of each modality, to have zero-mean and unit variance. Fig. 11 shows another tumor case in four different modality with different target label. We also present several example results on different methods in Fig. 12.

To demonstrate the performance on a dataset with real-world annotations, we have also evaluated our model on LIDC-IDRI. The "ground truth" labels in the experiments are generated by aggregating the multiple labels via Spatial STAPLE[14] as used in the curation of existing public datasets e.g., ISLES [10], MSSeg [11], Gleason'19 [12]. Fig. 13 presents several examples of segmentation results from different methods. We also measure the inter-reader consensus level by computing the IoU of annotations, and compare in Fig. 14 the estimates from our model against the values meansured on the real annotations. Furthermore, we divide the test dataset into low consensus (30% to 65%), middle consensus (65% to 75%) and high consensus (75% to 90%) subgroups and compare the performance in Fig. 15. Our method shows competitive ability to segment the challenging examples with low consensus values. Here we note that the consensus values in our test data range from 30% to 90%,, and compared the dice coefficient of our model with baselines.

On both BraTS and LIDC-IDRI dataset, our proposed model consistently achieves a higher dice similarity coefficient than STAPLE on both of the dense labels and single label scenarios (shown in Table. 4 and Table. 5). In addition, our model (with trace) outperforms STAPLE in terms of CM estimation by a large margin at $14.4\%$ on BraTS. In Fig. 11, we visualized the segmentation results on BraTS and the corresponding annotators' predictions. Fig. 12 presents four examples of the segmentation results and the corresponding annotators' predictions, as well as the baseline methods. As shown in both figures, our model successfully predicts the both the segmentation of lesions and the variations of each annotator in different cases.

| Models | BraTS DICE (%) | BraTS CM estimation | LIDC-IDRI DICE (%) | LIDC-IDRI CM estimation |
|---|---|---|---|---|
| Naive CNN on mean labels | $29.42 \pm 0.58$ | n/a | $56.72 \pm 0.61$ | n/a |
| Naive CNN on mode labels | $34.12 \pm 0.45$ | n/a | $58.64 \pm 0.47$ | n/a |
| Probabilistic U-net [24] | $40.53 \pm 0.75$ | n/a | $61.26 \pm 0.69$ | n/a |
| STAPLE [9] | $46.73 \pm 0.17$ | $0.2147 \pm 0.0103$ | $69.34 \pm 0.58$ | $0.0832 \pm 0.0043$ |
| Spatial STAPLE [14] | $47.31 \pm 0.21$ | $0.1871 \pm 0.0094$ | $70.92 \pm 0.18$ | $0.0746 \pm 0.0057$ |
| Ours with Global CMs | $47.33 \pm 0.28$ | $0.1673 \pm 0.1021$ | $70.94 \pm 0.19$ | $0.1386 \pm 0.0052$ |
| Ours without Trace | $49.03 \pm 0.34$ | $0.1569 \pm 0.0072$ | $71.25 \pm 0.12$ | $0.0482 \pm 0.0038$ |
| Ours | $\mathbf{53.47 \pm 0.24}$ | $\mathbf{0.1185 \pm 0.0056}$ | $\mathbf{74.12 \pm 0.19}$ | $\mathbf{0.0451 \pm 0.0025}$ |
| Oracle (Ours but with known CMs) | $67.13 \pm 0.14$ | $0.0843 \pm 0.0029$ | $79.41 \pm 0.17$ | $0.0381 \pm 0.0021$ |

Table 4: Comparison of segmentation accuracy and error of CM estimation for different methods trained with **dense labels** (mean $\pm$ standard deviation). The best results are shown in bald. Note that we count out the Oracle from the model ranking as it forms a theoretical upper-bound on the performance where true labels are known on the training data.

| Models | BraTS DICE (%) | BraTS CM estimation | LIDC-IDRI DICE (%) | LIDC-IDRI CM estimation |
|---|---|---|---|---|
| Naive CNN on mean & mode labels | $36.12 \pm 0.93$ | n/a | $48.36 \pm 0.79$ | n/a |
| STAPLE [9] | $38.74 \pm 0.85$ | $0.2956 \pm 0.1047$ | $57.32 \pm 0.87$ | $0.1715 \pm 0.0134$ |
| Spatial STAPLE [14] | $41.59 \pm 0.74$ | $0.2543 \pm 0.0867$ | $62.35 \pm 0.64$ | $0.1419 \pm 0.0207$ |
| Ours with Global CMs | $41.76 \pm 0.71$ | $0.2419 \pm 0.0829$ | $63.25 \pm 0.66$ | $0.1382 \pm 0.0175$ |
| Ours without Trace | $43.74 \pm 0.49$ | $0.1825 \pm 0.0724$ | $66.95 \pm 0.51$ | $0.0921 \pm 0.0167$ |
| Ours | $\mathbf{46.21 \pm 0.28}$ | $\mathbf{0.1576 \pm 0.0487}$ | $\mathbf{68.12 \pm 0.48}$ | $\mathbf{0.0587 \pm 0.0098}$ |

Table 5: Comparison of segmentation accuracy and error of CM estimation for different methods trained with only one label available per image (mean $\pm$ standard deviation). The best results are shown in bald.

Figure 11: The final segmentation of our model on BraTS and each annotator network predictions visualization. (Best viewed in colour: the target label is green.)

Figure 12: Visualisation of segmentation labels on BraTS dataset: (a) GT and simulated annotator's segmentations (Annotator 1 - 5); (b) the predictions from the supervised models.

| Input Image | Ground Truth | Annotator 1 label (Good Segmentation) | Annotator 2 label (Over Segmentation) | Annotator 3 label (Under Segmentation) | Annotator 4 label (Wrong Segmentation) | Naive CNN on Mean Labels | Naive CNN on Mode Labels | STAPLE | Spatial STAPLE | Our CNN | Our CNN (λ=0) | Oracle |

(a)                                                                                     (b)

Figure 13: Visualisation of segmentation labels on LIDC-IDRI dataset: (a) GT and simulated annotator's segmentations (Annotator 1 - 5); (b) the predictions from the supervised models.)

Figure 14: The consensus level amongst the estimated annotators is plotted against the ground truth on LIDC-IDRI dataset. The strong positive linear correlation shows that the variation in the inter-reader variability on different input examples (e.g., some examples are more ambiguous than others) is captured well. We do note, however, that the inter-reader variation seems more under-estimated for "easy" (i.e., higher consensus) samples.

Figure 15: Segmentation performance on 3 different subgroups of the LIDC-IDRI dataset with varying levels of inter-reader agreement. Our method shows *consistent* improvement over the baselines and the competing methods in all groups, showing its enhanced ability to segment challenging examples (i.e., low-consensus cases).

## A.4 Low-rank Approximation

Here we show our preliminery results on the employed low-rank approximation of confusion matrices for BraTS dataset, precluded in the main text. Table. 6 compares the performance of our method with the default implementation and the one with rank-1 approximation. We see that the low-rank approximation can halve the number of parameters in CMs and the number of floating-point-operations (FLOPs) in computing the annotator prediction while resonably retaining the performance on both segmentation and CM estimation. We note, however, the practical gain of this approximation in this task is limited since the number of classes is limited to 4 as indicated by the marginal reduction in the overall GPU usage for one example. We expect the gain to increase when the number of classes is larger as shown in Fig. 16.

| Rank | Dice | CM estimation | GPU Memory | No. Parameters | FLOPs |
|------|------|---------------|------------|----------------|-------|
| Default | $53.47 \pm 0.24$ | $0.1185 \pm 0.0056$ | 2.68GB | 589824 | 1032192 |
| rank 1 | $50.56 \pm 2.00$ | $0.1925 \pm 0.0314$ | 2.57GB | 294912 | 405504 |

Table 6: Comparison between the default implementation and low-rank (=1) approximation on BraTS. GPU memory consumption is estimated for the case with batch size = 1. Bot the total number of variables in the confusion matrices, and the number of FLOPs required in computing the annotator predictions.

Lastly, we also describe the details of the devised low-rank approximation. Analogous to Chandra and Kokkinos's work [48] where they employed a similar approximation for estimating the pairwise terms in densely connected CRF, we parametrise the spatial CM, $\hat{\mathbf{A}}_\phi^{(r)}(\mathbf{x}) = \mathbf{B}_{1,\phi}^{(r)}(\mathbf{x}) \cdot \mathbf{B}_{2,\phi}^{T,(r)}(\mathbf{x})$ as a product of two smaller rectangular matrices $\mathbf{B}_{1,\phi}^{(r)}$ and $\mathbf{B}_{2,\phi}^{(r)}$ of size $W \times H \times L \times l$ where $l << L$. In this case, the annotator network outputs $\mathbf{B}_{1,\phi}^{(r)}$ and $\mathbf{B}_{2,\phi}^{(r)}$ for each annotator in lieu of the full CM. Two separate rectangular matrices are used here since the confusion matrices are not necessarily symmetric. Such low-rank approximation reduces the total number of variables to $2WHLl$ from $WHL^2$ and the number of floating-point operations (FLOPs) to $WH(4L(l-0.25)-l)$ from $WH(2L-1)L$. Fig. 16 shows that the time and space complexity of the default method grow quadratically in the number of classes while the low-rank approximations have linear growth.

Figure 16: Comparison of time and space complexity between the default implementation and the low-rank counterparts. (a) compares the number of parameters in the confusion matrices while (b) shows the number of FLOPs required to compute the annotator predictions (the product between the confusion matrices and the estimated true segmentation probabilities).

# B  Implementation details

Our method is implemented in Pytorch [49]. Our network is based on a 4 down-sampling stages 2D U-net [50], the channel numbers for each encoders are 32, 64, 128, 256, we also replaced the batch normalisation layers with instance normalisation. Our segmentation network and annotator network share the same parameters apart from the last layer in the decoder of U-net, essentially, the overall architecture is implemented as an U-net with multiple output last layers: one for prediction of true segmentation; others for predictions of noisy segmentation respectively. For segmentation network, the output of the last layer has c channels where c is the number of classes. On the other hand, for annotator network, by default, the output of the last layer has $L \times L$ number of channels for estimating confusion matrices at each spatial location; when low-rank approximation is used, the output of the last layer has $2 \times L \times l$ number of channels. The Probabilistic U-net implementation is adopted from `https://github.com/stefanknegt/Probabilistic-Unet-Pytorch`, for fair comparison, we adjusted the number of the channels and the depth of the U-net backbone in Probabilistic U-net to match with our networks. All of the models were trained on a NVIDIA RTX 208 for at least 3 times with different random initialisations to compute the mean performance and its standard deviation (run 3 times of the experiments with the same initialization). The Adam [51] optimiser was used in all experiments with the default hyper-parameter settings. We also provide all of the hyper-parameters of the experiments for each data set in Table 7. We also kept the training details the same between the baselines and our method.

| Data set | Learning Rate | Epoch | Batch Size | Augmentation | weight for regularisation ($\lambda$) |
|---|---|---|---|---|---|
| MNIST | 1e-4 | 60 | 2 | Random flip | 0.7 |
| MS | 1e-4 | 55 | 2 | Random flip | 0.7 |
| BraTS | 1e-4 | 60 | 8 | Random flip | 1.5 |
| LIDC | 1e-4 | 75 | 4 | Random flip | 0.9 |

Table 7: Hyper-parameters used for respective datasets.

## B.1  Pytorch implementation of loss function

The following is the Pytorch implementation of the loss function in eq. (4). We also intend to clean up the whole codebase and release in the final version.

```python
import torch
import torch.nn as nn

def loss_function(p, cms, ts, alpha):
    """
    Args:
        p (torch.tensor): unnormalised probabilities from the segmentation network
            of size (batch, num classes, height, width)
        cms (list of torch.tensors): a list of estimated unnormalised (but positive)
            confusion matrices  from the annotator network, each with size
            (batch, num classes, num classes, height, width)
        ts (list of torch.tensors): a list of segmentation labels from noisy annotators,
            each with size (batch, num classes, height, width)
        alpha (float): weight for the trace regularisation
    """
    main_loss = 0.0
    regularisation = 0.0
    b, c, h, w = p.size()  # b: batch size; c: class number, h: height, w: width
    p = nn.Softmax(dim=1)(p)

    # reshape p: [b, c, h, w] => [b*h*w, c, 1]
    p = p.view(b, c, h*w).permute(0, 2, 1).contiguous()
    p = p.view(b*h*w, c, 1)

    # iterate over the confusion matrices & noisy labels from different annotators
    for j, (cm, t) in enumerate(zip(cms, ts)):
        # cm: confusion matrix of noisy annotator j
        # t: label for noisy segmentation of noisy annotator j
        # reshape cm: [b, c, c, h, w]=> [b*h*w, c, c]
        cm = cm.view(b, c**2, h * w).permute(0, 2, 1).contiguous()
        cm = cm.view(b*h*w, c**2).view(b*h*w, c, c)
        cm = cm / cm.sum(1, keepdim=True) # normalise the confusion matrix along columns
        # compute the estimated annotator's noisy segmentation probability
        p_n = torch.bmm(cm, p).view(b*h*w, c)
        # reshape p_n: [b*h*w, c , 1] => [b, c, h, w]
        p_n = p_n.view(b, h*w, c).permute(0, 2, 1).contiguous().view(b, c, h, w)
        # calculate the pixelwise cross entripy loss
        main_loss += nn.CrossEntropyLoss(reduction='mean')(p_n, t.view(b, h, w).long())
        # calculate the mean trace
        regularisation =  torch.trace(torch.sum(cm, dim=0)).sum() / (b*h*w)

    regularisation = alpha*regularisation
    return main_loss + regularisation
```

# C Proof of Theorem 1

We first show a specific case of Theorem 1 when there is only a single annotator, and subsequently extend it to the scenario with multiple annotators. Without loss of generality, we show the result for an arbitrary choice of a pixel in a given input image $\mathbf{x} \in \mathbb{R}^{W \times H \times C}$. Specifically, let us denote the estimated confusion matrix (CM) of the annotator at the $(i,j)^{\text{th}}$ pixel by $\hat{\mathbf{A}} := [\hat{\mathbf{A}}_{\phi}(\mathbf{x})_{ij}] \in [0,1]^{L \times L}$, and suppose the true class of this pixel is $k \in [1,...,L]$ i.e., $\mathbf{p}(\mathbf{x}) = \mathbf{e}_k$ where $\mathbf{e}_k$ denotes the $k^{\text{th}}$ elementary basis. Let $\hat{\mathbf{p}}_{\theta}(\mathbf{x})$ denote the $L$-dimensional estimated label distribution at the corresponding pixel (instead of over all the whole image).

**Lemma 1.** *If the annotator's segmentation probability is fully captured by the model for the $(i,j)^{th}$ pixel in image $\mathbf{x}$ i.e., $\hat{\mathbf{A}} \cdot \hat{\boldsymbol{p}}_{\theta}(\boldsymbol{x}) = \mathbf{A} \cdot \boldsymbol{p}(\boldsymbol{x})$, and both $\hat{\mathbf{A}}, \mathbf{A}$ satisfy that $a_{kk} > a_{kj}$ for $j \neq k$ and $\hat{a}_{ii} > \hat{a}_{ij}$ for all $i,j$ such that $j \neq i$, then $tr(\hat{\mathbf{A}})$ is minimised when $\hat{\mathbf{A}} = \mathbf{A}$. Furthermore, if $tr(\hat{\mathbf{A}}) = tr(\mathbf{A})$, then the true label is fully recovererd i.e., $\hat{\boldsymbol{p}}_{\theta}(\boldsymbol{x}) = \boldsymbol{p}(\boldsymbol{x})$ and the $k^{th}$ column in $\hat{\mathbf{A}}$, $\mathbf{A}$ are the same.*

*Proof.* We first show that the $k^{\text{th}}$ diagonal element in $\mathbf{A}$ is smaller than or equal to its estimate in $\hat{\mathbf{A}}$. Since $\mathbf{p}(\mathbf{x}) = \mathbf{e}_k$ is a one-hot vector, $\hat{\mathbf{A}} \cdot \hat{\mathbf{p}}_{\theta}(\mathbf{x}) = \mathbf{A} \cdot \mathbf{p}(\mathbf{x})$ holds and $\hat{a}_{kk} > \hat{a}_{kj} \forall j \neq k$, it follows that:

$$a_{kk} = \left\langle [\hat{a}_{k1},...,\hat{a}_{kL}],\ \hat{\mathbf{p}}_{\theta}(\mathbf{x}) \right\rangle \tag{5}$$

$$\leq \left\langle [\hat{a}_{kk},...,\hat{a}_{kk}],\ \hat{\mathbf{p}}_{\theta}(\mathbf{x}) \right\rangle = \hat{a}_{kk}. \tag{6}$$

The possibility of equality in the above comes from the fact that all entries in $\hat{\mathbf{p}}_{\theta}(\mathbf{x})$ except the $k$th element could be zeros. Now, the assumption that there is a single ground truth label $k$ for the $(i,j)^{\text{th}}$ pixel means that all the values of the true CM, $\mathbf{A}$ are uniformly equal to $1/L$ except the $k^{\text{th}}$ column. In addition, since the diagonal dominance of the estimated CM means each $\hat{a}_{ii}$ is at least $1/L$, we have that

$$\text{tr}(\mathbf{A}) = \frac{L-1}{L} + a_{kk} \leq \sum_{j \neq k} \hat{a}_{jk} + \hat{a}_{kk} = \text{tr}(\hat{\mathbf{A}}).$$

It therefore follows that when $\hat{\mathbf{A}} = \mathbf{A}$ holds, the trace of $\text{tr}(\hat{\mathbf{A}})$ is the smallest. Now, we show that when this holds i.e., $\text{tr}(\mathbf{A}) = \text{tr}(\hat{\mathbf{A}})$, then the $k^{\text{th}}$ columns of the two matrices match up.

By way of contradiction, let us assume that there exists a class $k' \neq k$ for which the estimated label probability is non-zero i.e., $\hat{p}_{k'} := [\hat{\mathbf{p}}_{\theta}(\mathbf{x})]_{k'} > 0$. This implies that $1 - \hat{p}_k > 0$. From eq. (6), if the trace of $\mathbf{A}$ and $\hat{\mathbf{A}}$ are the same, then $a_{kk} = \hat{a}_{kk}$ also holds and thus we have $\hat{a}_{kk} = \sum_j \hat{a}_{kj} \hat{p}_j$. By rearranging this equality and dividing both sides by $1 - \hat{p}_k$, we obtain $\hat{a}_{kk} = \sum_{j \neq k} \frac{\hat{p}_j}{1 - \hat{p}_k} \hat{a}_{kj}$. Now, as we have $\hat{a}_{kk} > \hat{a}_{kj}, j \neq k$, it follows that

$$\hat{a}_{kk} < \hat{a}_{kk} \sum_{j \neq k} \frac{\hat{p}_j}{1 - \hat{p}_k} = \hat{a}_{kk}$$

which is false. Therefore, the trace quality implies $\hat{p}_k = 1$ and thus from $\hat{\mathbf{A}} \cdot \hat{\mathbf{p}}_{\theta}(\mathbf{x}) = \mathbf{A} \cdot \mathbf{p}(\mathbf{x})$, we conclude that the $k^{\text{th}}$ columns of $\hat{\mathbf{A}}$ and $\mathbf{A}$ are the same.

$\square$

We note that the equivalent result for the expectation of the annotator's CM over the data population was provided in [52] and [28]. The main difference is, as described in the main text, that we show a slightly weaker version of their result in a sample-specific scenario.

Now, we show that the main theorem follows naturally from the above lemma. As a reminder, we recite the theorem below.

**Theorem 1.** *For the $(i,j)^{th}$ pixel in a given image* **x**, *we define the mean confusion matrix (CM)* $\boldsymbol{A}^* := \sum_{r=1}^{R} \pi_r \hat{\boldsymbol{A}}^{(r)}$ *and its estimate* $\hat{\boldsymbol{A}}^* := \sum_{r=1}^{R} \pi_r \hat{\boldsymbol{A}}^{(r)}$ *where* $\pi_r \in [0,1]$ *is the probability that the annotator $r$ labels image* **x**. *If the annotator's segmentation probabilities are perfectly modelled by the model for the given image i.e.,* $\hat{\boldsymbol{A}}^{(r)}\hat{\boldsymbol{p}}_\theta(\boldsymbol{x}) = \boldsymbol{A}^{(r)}\boldsymbol{p}(\boldsymbol{x}) \forall r = 1,...,R$, *and the average true confusion matrix $\boldsymbol{A}^*$ at a given pixel and its estimate $\hat{\boldsymbol{A}}^*$ satisfy that $a^*_{kk} > a^*_{kj}$ for $j \neq k$ and $\hat{a}^*_{ii} > \hat{a}^*_{ij}$ for all $i,j$ such that $j \neq i$, then $\boldsymbol{A}^{(1)},...,\boldsymbol{A}^{(R)} = argmin_{\hat{\boldsymbol{A}}^{(1)},...,\hat{\boldsymbol{A}}^{(R)}} \left[ tr(\hat{\boldsymbol{A}}^*) \right]$ and such solutions are **unique** in the $k^{th}$ columns where $k$ is the correct pixel class.*

*Proof.* A direct application of Lemma 1 shows firstly that $tr(\hat{\boldsymbol{A}}^*)$ is minimised when $\hat{\boldsymbol{A}}^{(r)} = \boldsymbol{A}^{(r)}$ for all $r = 1,...,R$ (since that ensures $\boldsymbol{A}^* = \hat{\boldsymbol{A}}^*$). Secondly, it implies that minimising $tr(\hat{\boldsymbol{A}}^*)$ yields $\hat{\mathbf{p}}_\theta(\mathbf{x}) = \mathbf{p}(\mathbf{x})$. Because we assume that annotators' noisy labels are correctly modelled i.e., $\hat{\mathbf{A}}^{(r)}\hat{\mathbf{p}}_\theta(\mathbf{x}) = \mathbf{A}^{(r)}\mathbf{p}(\mathbf{x}) \forall r = 1,...,R$, it therefore follows that the $k^{th}$ column in $\hat{\mathbf{A}}^{(r)}$ and $\mathbf{A}^{(r)}$ are the same.

$\square$