[Reviews · NeurIPS 2020]

Review 1

Summary and Contributions: The authors develop a model for learning reliability of individuals from noisy data, and to show the true segmentation label distributions. They propose two CNNs; one as an 'annotator' that must be as bad as possible while remaining as close as possible to training data. The second to provide an automatic segmentation which is combined with a matrix of each annotator. The output should be a segmentation that is as close as possible to the 'true segmentation' rather than just being an average of multiple segmentations for a given image.

Strengths: The motivation that we lack a good ground truth for segmentation is strong. This is indeed a challenge and the discussion of prior work here is basically good. The use of three different metrics to evaluate is helpful.

Weaknesses: If I understand correctly, STAPLE (reference 9) just aggregates the ground truths of multiple different annotators. This is different from automatic segmentation, which is what the authors are looking at with the other baselines (their model, naive CNNs and probabilistic U-net). So it’s somewhat confusing to include this in the table as if it was an automatic segmentation benchmark. In general the comparison the benchmarks on segmentation accuracy is confusing, as it is difficult on a first read to understand if the goal is overall accuracy - i.e., the proposed model is a way of improving accuracy for automatic segmentation on previously unseen images - or if the goal is modelling the reliability of individual labellers, and that this information could be used in education, or as a filter to identify just the GT annotations to use for model training. The authors should make sure this is more clear. The work assumes all individuals err in the a similar way, but when discussing future work the authors do mention they want to include metadata that might help learn deviations in how humans make mistakes. An important piece of metadata is whether the individuals have experiencing segmenting professionally (e.g., radiotherapy) or if they have just done it for a study. I.e., errors from the action of segmenting itself will be different from errors in the interpretation of medical imaging boundaries. UPDATE: the authors rebuttal addressed my concerns sufficiently to warrant an upgraded assessment.

Correctness: The claims are correct. The claim that there is only one true segmentation map of the underlying anatomy is not always correct in medical imaging. While I agree with the authors that multiple hypotheses can arise from ambiguities in human interpretations, ambiguities in the imaging process itself may mean that multiple variations of a ground truth are plausible.

Clarity: While other figures were displayed in colour, Figure 2 was black and while. This is a shame as the legend even stated that colours were needed!

Relation to Prior Work: Prior work is clearly discussed.

Reproducibility: Yes

Additional Feedback:


Review 2

Summary and Contributions: The segmentation labels provided by multiple human annotators for a given training dataset are potentially noisy versions of the real, unknown ground-truth. Authors explore this idea to propose a method that simultaneously learns to estimate the pixel-wise confusion matrices (CMs) for each annotator and the real ground-truth. Training minimizes the cross-entropy between the noisy labels of each annotator and the estimated labels for that annotator obtained by multiplying the estimated CMs and estimated labels. An additional regularization term minimizes the reliability of each annotator. Authors prove that, under some mild conditions, this regularization produces CMs whose column k-th will match the column k-th of the true CMs, being k the correct class. Experiments apply this idea to train segmentation networks by aggregating information from five different noisy annotators. The attained results outperforms other baselines methods, sometimes by a large margin.

Strengths: To the extent of my knowledge, this is the first work that learns how to aggregate the information from multiple annotators by modeling the nature of their mistakes while training a segmentation network. Unlike previous works, authors model annotation errors conditioned not only to the correct class, but also to the contents of the input image in a pixel-wise manner. These is a sensible idea, as annotators are prone to making mistakes in an image-dependent manner, and they are unlikely to consistently make the same type of mistakes throughout the entire image. The proposed regularization term is critical for the proper operation of the training. Authors show its importance both theoretically in Theorem 1 and empirically in the experimental section. Finally, I found the conducted experiments extensive and convincing. Authors compare their method against a large collection of baselines in four different dataset with both synthetic and real annotators. The experiments are limited to medical datasets, but I do not find this to be a limitation, as medical datasets are challenging enough to be representative of the performance in other real-world scenarios.

Weaknesses: The method and theoretical contributions are very close to those from [28]. [28] proposes to learn a global CM for each annotator in classification tasks. The present work extends this idea for segmentation tasks by conditioning the CMs to the input image via a fully convolutional network. While this is not a trivial improvement, I find the way it is done to be a straightforward extension from [28]. [28] also introduces the trace of the CMs as a regularization term, and proves that it leads to the true CMs under some mild conditions. Theorem 1 and its proof are minor modifications of these theoretical contributions from[28] to the case of pixel-wise CMs. Given the similarity to [28], I am surprised that authors did not include a "global CMs" method (in the spirit of [28] but in a segmentation task) as a baseline in the experimental section. I think it is critical that authors show that global CMs underperform when compared to the pixel-wise image-conditioned CMs of the proposed method. This is not a trivial point to me: while global CMs might ignore important image information, they are also much less prone to overfitting than local CMs.

Correctness: Both the claims and the method seem correct to me. I did not find issues regarding this point.

Clarity: Sections 3 and 4 have multiple issues. In some cases I had to guess what the authors did. In some other cases I had to resort to [28] to solve my doubts. Section 3 has some notation typos and issues that hinder the readability: - There are a considerable and sometimes confusing abuse of notation regarding per-pixel operations. This is specially problematic in Equations (3) and (4). For example, the dot product A(x)·p(x) is in practice multiplying two multichannel images (one containing a probability distribution per pixel and one containing a CM per pixel). The CE operator also acts over entire images, but the reader needs to guess how it aggregates the per-pixel cross-entropy into a single value. It is not clear to me how to interpret 1(y_n^{(r)} \in \mathcal{S}(x)), given that y_n^{(r)} is a vector of labels (one label per pixel) and \mathcal{S}(x) seems to be a set of annotators. - Is there a different between S(x_n) introduced in line 128 and the \mathcal{S}(x_n) described in line 181? I guess they are the same function. But in that case, the term 1(y_n^{(r)} \in \mathcal{S}(x)) in equations (3) and (4) should be 1(r in \mathcal{S}(x)), as happens in line 168. - In Equation (4) x_i should be x_n. - In line 195, the mean CM A^* and its estimate \hat{A}^* have the very same definition. - Is the definition of \hat{A}^{(r)} in line 194 correct? Section 4 also has multiple issues: - Tables 1 and 2 show results for "dense labels" and "one label per image", respectively. However, I had to read [28] to understand what "one label per image" means and how that label is chosen. I could not find this explanation in the paper. - Tables 1, 2, and 3 show results with standard deviations, but it is not specified over how many runs. - It is not clear to me how the "Oracle (Ours but with known CMs)" of Table 1 was performed. - The caption of Figure 2 ("CMs of 5 simulated annotators on MNIST dataset (Best viewed in colour [...])") is unrelated to its contents. I do not see CMs in Figure 2. There are no colors. It shows results for MNIST and MS Lesion.

Relation to Prior Work: The review seems correct to me. I do not have objections regarding this point.

Reproducibility: Yes

Additional Feedback:


Review 3

Summary and Contributions: The main goal of this paper is to estimate the unknown true segmentation map. To solve the high inter-observer variability problem in the medical images segmentation task, the author proposed a supervised disentangling model trained by multiple noisy annotations. The author used the probabilistic model to describe the principle of noisy medical annotations. The theory is clear and logical. One component of the model was trained to learn the real segmentation map, and the other models the annotators' behavior to reconstruct noisy annotations based on it. The first component was trained to generate a realistic map by minimizing the cross-entropy of noisy annotators. This paper demonstrated the method on four datasets, including two medical datasets with synthetic annotations and real annotations by multiple experts. Compared to five label fusion baselines, this method gets better performance on each criterion and gets better results visually.

Strengths: This paper describes a common problem in medical image segmentation, and it's meaningful to solve it. In summary, the proposed framework is innovative. The strengths are given as follows: 1. The task is beneficial for the application of machine learning in medical images. Since the high inter-observer variability is ubiquitous in this field, noisy labeling problems might restrict many methods' performance. 2. Compared to other methods, using probabilistic models to combine the ground truth map and annotators' characteristics is closer to reality. It's brilliant to use two parts to simulate the generation process of several noisy annotations, which guides the model to learn the real segmentation map. 3. The related work is elaborate. The author introduced various fusing label methods and analyzed the features and deficiency of previous researches. 4. The segmentation results are better than other related methods on both digit images and medical images, and the generated segmentation map is visually accurate merely by several noisy annotations. 5. The author uses public datasets to evaluate and give enough details of this work to be reproducible.

Weaknesses: 1. The probability model is impressive, but the second assumption might limit the model performance. In practice, annotations of different pixels aren't independent. The pixels around a region are more likely to be labeled the same category than pixels far away. It's more reasonable to integrate each point's surrounding pixels in GT map to get macroscopic information. 2. The method was validated on four datasets, but only one dataset has real-world annotations. The morphological operations to generate noisy labels seem to be simplex, and the annotations in the LIDC-IDRI dataset have little variability. The author mentioned the Gleason'19 dataset, which has high inter-observer variability. It would be more credible to evaluate on datasets like that.

Correctness: Yes

Clarity: Yes. The paper is written clearly.

Relation to Prior Work: Yes. This author pointed out the features of former researches on fusing labels. This work solves the common drawback by integrating information across multiple annotations. So the differences between this work and previous work are clearly indicated.

Reproducibility: Yes

Additional Feedback: 1. Since the annotations of the BraTS and the LIDC-IDRI dataset have few differences, it’s better to adjust the figures to be more intuitive. Besides, the annotations of the first and third examples in the LIDC-IDRI dataset are too similar, which can’t embody the performance. 2. The title is Disentangling Human Error from Ground Truth in Segmentation. But in the experiments part, it’s more like correcting boundary deviation than disentangling error. The inverse segmentation as an error in the BraTS dataset is not convincing enough. 3. As mentioned earlier, I think integrating surrounding information in the probability model will help learn the realistic pattern. Take brain segmentation, for example. Some experts may draw a large circle to point out the problem region, even though there are some small parts of the normal area. And some experts may be precise and point out every single region in the slide. And this phenomenon can be modeled by a probability framework with macroscopic information. ---------------------------------------------------- After reading the rebuttal, the work can be summarized as follows: -- The authors propose a novel segmentation approach to estimate the latent ground truth from several noisy annotations. -- The main idea is using predicted confusing matrixes to model the annotators, which assist the network in reconstructing the ground truth. --The whole framework is like an extension of reference [28] from classification to segmentation, but the image-conditioned CMs proposed in this paper are more reasonable than global CMs, especially in medical images tasks. (weakness) --Besides, the authors did extensive experiments in four datasets to prove the performance of the model. And the supplementary experiment result also shows the superiority of image-dependent CMs. However, maybe there are few satisfactory datasets, but it will be more convincing to validate real-world datasets with high variation. (weakness) I will hold my original score.


Review 4

Summary and Contributions: The authors of this paper propose a method that jointly learns the ground-truth segmentation labels and the residual errors incurred by human annotators. There are two major contributions. First, the proposed model is able to learn both tasks without access to the groud-truth labels. Second, the model is able to learn the noise pattern associated with each annotator on a population rather than on a single image.

Strengths: First of all, noise among labels is a very common problem in the medical imaging domain. The problem that this paper is trying to solve is very clinically relevant. Moreover, the authors propose a rigorous framework to formulate the task. The design of the trace-based regularization is particularly novel. In addition, extensive experiments are performed to demonstrate the superior performance of the model.

Weaknesses: The major weakness of this paper is the lack of experiments with non-synthetic labeler noise. While the proposed method is benchmarked on the real-world dataset, all the noisy labels are created by artificially distorting the segmentation labels. However, the synthetic noise generation process does not necessarily reflect the noise generated by human annotators. For instance, annotator 2,3,5 impose very severe distortion that a professional annotator, especially in medical imaging, wouldn't do. To truly prove the clinical usage of this paper, the author needs to perform experiments on the settings where noise is generated by human labelers. An ideal dataset would contain GT labels created by experienced annotators (radiologists for instance) and noise labels created by inexperienced annotators (medical students). Update: this concern is mostly addressed in the author feedback.

Correctness: The experiment set up is sensible. The authors provide sufficient support for the claims made in this paper.

Clarity: The overall quality of writting in this paper is satisfactory. The authors could improve the quality of writting by avoid complicated long sentences. There are a few minor grammar error / typo such as line 24, 226.

Relation to Prior Work: Yes, there is suffcient discussion on the prior works as well as how the proposed method is different from them.

Reproducibility: Yes

Additional Feedback:

[Author Response · NeurIPS 2020]

We are grateful for all the reviewers for their constructive feedback, which will undoubtedly improve the quality of the manuscript. As four reviewers acknowledge, we present a novel segmentation approach to learning robustly in the presence of large expert disagreement in annotations, and demonstrate its utility in a range of datasets from medical imaging where such problem is particularly pertinent. Our work is the first instance of an end-to-end supervised segmentation method that simultaneously estimates the reliability of individual annotators and the true segmentation distribution from noisy observations alone. We will open-source the code and label simulator upon publication.

The main criticisms from the reviews are summarised as:

(1) **R1.3.1:** Relevance of STAPLE and its variants as baselines is unclear
(2) **R1.3.2:** The overall goal needs clarification: is it segmentation accuracy or the utility of modeling the reliability of annotators in downstream tasks?
(3) **R2.3.1:** The method and theoretical contributions are very close to those from [28].
(4) **R2.3.2:** Need to include the *"global"* confusion matrix (CM) based model in the performance comparison.
(5) **R3.3.1:** Assuming annotations of different pixels are independent might limit the model performance.
(6) **R3.3.2:** The LIDC-IDRI dataset has little inter-reader variability, and may be more credible to test on a dataset with higher variability e.g., Gleason'19.
(7) **R4.3.1:** All annotations are simulated, and the method should be evaluated on a dataset with real noisy annotations.

**Reg. (1)**, STAPLE is the most prevalent label fusion framework used in the curation of training datasets for medical image segmentation (including many public benchmarks), where the initial labels are typically noisy. The advancement made in our work can also be viewed as a translation of the STAPLE framework to the supervised learning setting. Thus it is necessary to know how much improvement the learning based method achieved compared to such label fusion methods.

**Reg. (2)**, our primary goal is to learn to segment unseen images accurately when the annotations in the training data are very noisy. We attain this by modelling explicitly the reliability of the individual annotators, and as R1 highlights, such information can be potentially used in downstream applications e.g., education and active label acquisition. However, this remains future work, and to clarify, we have removed the last sentence of the abstract and expand on the future outlook in discussion.

**Reg. (3)**, while the work [28] builds on their conceptual framework, extending to segmentation demands substantial technical differences. Firstly, to capture the correlations involved in image segmentation, the method yields the estimate of annotator reliability in every pixel, which is crucial in capturing the complex spatial variations in annotators' characteristics, and is absent in [28] which only addressed image classifications. Secondly, we modelled the reliability of annotators as a function of the input image in a stark contrast with [28] that merely estimated the population average as additional parameters — this enables estimating the annotator's reliability on a per example basis. Lastly, the theoretical justification for the trace regularisation was extended from the "global" confusion matrix scenario to the "local" setting—this is not trivial and we could only prove a weaker (but sufficient) statement that the relevant column of the CMs is recoverable but not the entire matrices. We will state these differences more clearly in intro and section 3.

**Reg. (4)**, we agree with R2 that it is critical to include such a baseline to assess the benefits of modelling CMs as a function of the image—thank you for the suggestion. We have implemented this and included its performance in all 8 quantitative comparisons. While performing better than mean/mode label baselines, the table below shows the global CM model performs consistently worse than our model based on the image-dependent pixel-wise CMs with differences indicated in red:

| Dataset | DICE (dense labels) | CM estimation, mse (dense labels) | DICE score (1 label per image) | CM estimation, mse (1 label per image) |
|---|---|---|---|---|
| MNIST | $79.21 \pm 0.41 \, (-3.71)$ | $0.1132 \pm 0.0028 \, (+0.0239)$ | $59.01 \pm 0.65 \, (-17.47)$ | $0.1953 \pm 0.0041 \, (+0.0624)$ |
| MS | $61.58 \pm 0.59 \, (-5.97)$ | $0.1449 \pm 0.0051 \, (+0.0638)$ | $40.32 \pm 0.68 \, (-16.11)$ | $0.1974 \pm 0.0063 \, (+0.0432)$ |
| BraTS | $47.33 \pm 0.28 \, (-6.14)$ | $0.1673 \pm 0.1021 \, (+0.0488)$ | $41.76 \pm 0.71 \, (-4.45)$ | $0.2419 \pm 0.0829 \, (+0.0843)$ |
| LIDC-IDRI | $70.94 \pm 0.19 \, (-3.18)$ | $0.1386 \pm 0.0052 \, (+0.0935)$ | $63.25 \pm 0.66 \, (-4.87)$ | $0.1382 \pm 0.0175 \, (+0.0795)$ |

**Reg. (5)**, we agree with R3 that independence assumption between pixels in annotation might limit the performance. We note, however, that the annotations are assumed to be only conditionally independent between pixels given the input image, and thus the model can still capture some correlations in the output segmentation labels that are explained in the input image. We additionally note that such independence assumption is typically made in most of the deep learning based segmentation methods, and thus a posprocessing method such as Gaussian CRF is commonly used to capture the missed correlations. We believe the same problem applies to the annotation modelling—we note this limitation and mention such correlation modelling as future work in the discussion.

**Reg. (6)**, the LIDC-IDRI dataset contains some cases with very high inter-reader variability, and we will clarify in section 4. For example, Fig.14 in the supplementary material shows that many examples have low consensus levels in the range of 30%-50% mIoU between annotators. The third row in Fig.7 also shows one such example where Annotator 4 completely misses the abnormality. We believe that our initial promising results motivate us to test the method on an even more challenging dataset such as Gleason'19 in the future. However, we do note that Gleason'19 does not have curated ground-truth and resort to a label fusion method to create "gold standard" labels. An inspection with our clinical collaborators has revealed that many of such "gold standard" are not realistic, implying that an extra care is needed.

**Reg. (7)**, we have in fact evaluated our method with real annotations—the LIDC-IDRI dataset contains annotations per input from 4 different radiologists. Unlike the MNIST, MS and Brats datasets with synthetic noisy labels, we use LIDC-IDRI to evaluate the utility of our work in the presence of real-world noisy labels. We will clarify in Section 4.

[Meta-Review · NeurIPS 2020]

The paper proposes an approach to model annotator errors to come up with a consensus annotation from multiple ones. It is essentially an extension of [28] from classification to segmentation. The authors claim this is an extensive change in setting, and the reviewers in the main agree that it is sufficient. The reviewers unanimously agree that the paper is at least marginally above the acceptance threshold.